# The Digitalization Transformation of Commercial Banks and Its Impact on Sustainable Efficiency Improvements through Investment in Science and Technology

**Lihua Zuo** [1,*]**, Jack Strauss** [2,*] **and Lijuan Zuo** [1]

1   School of Economics and Management, Beijing Jiaotong University, Beijing 100044, China; 18113089@bjtu.edu.cn
2   Reiman School of Finance, University of Denver, 2101 S. University Blvd, Denver, CO 80208, USA
*   Correspondence: 16113106@bjtu.edu.cn (L.Z.); jack.strauss@du.edu (J.S.)

**Abstract:** The COVID-19 epidemic has accelerated the digital economy's pervasiveness throughout the Chinese economy, leading to a sharp rise in demand for "contactless" services in the financial industry. We examine the digital transformation of the Chinese banking industry using the DEA-Malmquist index method, supplemented by a distance function and time to compare the dynamic changes of productivity. Our paper then conducts an empirical study on the digital transformation of Chinese commercial banks based on their improvements in efficiency. We analyze banks with superior efficiency in science and technology investment and evaluate their digital maturity and digital transformation experience. Results show that digitalization investment has contributed to substantial production efficiency improvement for commercial banks; however, heterogeneity exists across banks. We further advocate a path for banks' digital transformation based on theoretical research and empirical digital transformation experience in this area.

**Keywords:** commercial banks; financial technology; input efficiency DEA-Malmquist model; digital transformation

## 1. Introduction

In recent decades, commercial banks in China have invested heavily in science and technology. These investments have led to the development of financial technology (fintech) and have substantially altered commercial bank performance. Fintech uses technology to provide innovative financial services [1] and promotes the innovation of the financial system through productivity advances and reform of technical tools. Fintech is currently being applied to create new business models, novel applications and processes, and innovative products, which markedly impact financial markets, institutions, and services. Investment in fintech includes bank's payment and clearing systems, e-money, online lending, big data, blockchain, cloud computing, artificial intelligence, intelligent investment consultant, intelligent contract and other fields. Large investments in these arenas are transforming core banking, insurance and payment processes, and total production efficiency. Our paper establishes a theoretical framework to assess the channels through which commercial bank's investment in fintech and digitalization affects sustainable efficiency improvements. Our empirical analysis demonstrates that fintech and digitalization investment generate sustainable bank technology improvements.

The development of fintech's impact on commercial banks can be divided into three stages. Stage one includes the introduction of IT. By applying traditional IT hardware and software at this stage, the commercial banks in the financial industry realize the automation of office management and business operation and improve business efficiency. Stage two is the introduction of internet finance. Commercial banks build online business platforms, attract customers and information through the Internet or mobile terminal channels, and

realize the interconnection of transaction, payment, and capital flow in financial business. Stage three applies fintech development. Banks begin to adopt new IT technologies for financial information collection, risk pricing models, investment decision-making processes, and credit intermediaries. The extant literature on fintech finds that machine learning algorithms in credit decisions improve efficiency [2]. Therefore, fintech enhances the efficiency of commercial banks through big data credit surveys, intelligent investment consultants, supply chain finance, etc.

Fintech development generates financial innovations for commercial banks [3]. From a global perspective, investment for fintech has been rapidly rising [4] and now enables entrepreneurs in banks to face potential market challenges [5]. Fintech is also presently enhancing inclusive finance, boosting risk control capabilities, and improving the total productivity efficiency of commercial banks.

Fintech significantly alters the financial landscape through commercial banks' investment in science and technology, especially in the post-COVID era. China's digital economy is rapidly transforming. In 2019, the digital economy's value-added reached 35.8 trillion yuan, accounting for 36.2% of GDP and 67.7% of GDP growth [6]. The COVID-19 epidemic has further markedly accelerated demand for "contactless" services in the financial industry and the rapid development of AI technology, big data, cloud computing, blockchain, and other new technologies. To exploit the digital economy's revolution, commercial banks should consider how to integrate into the new digital economy with innovative formats and models and seize the opportunity for a new round of technological [7] upgrading and industrial transformation.

Boole believed that digitization was a combination process of signals, sounds, images, and objects represented by digital symbols. More recently, Sjodin [8] wrote that digitization is a social upgrading led by using advanced technologies, such as the Internet, big data, social media, blockchain, and digital currency. The digitalization process and its subsequent changes are called digital transformation [9,10]. Rogers [11] posited that this transformation is mainly reflected in five aspects: customer, data, innovation, value, and competition. Zhou [12] and others defined digitization as a digital process of building a physical world based on digital technology. These digital advances use artificial intelligence, cloud computing, and other core technologies and can restructure the firm and its talent culture and substantially contribute to innovations.

Most of the existing research on digital transformation explores regional and industrial level issues [13] and underscores the significant positive impact of the development of the Internet on upgrading the financial industry. At the firm level, research emphasizes tax burdens [14] from the perspective of external factors, including financing support [15]; other works focus on the labor flow from the perspective of internal factors and boosting the path of digitalization to transform enterprises [16].

This paper starts from enterprise-level research on the theory of financial-technology investment and analyzes the digital transformation strategies of commercial banks. Innovation theory and capital allocation efficiency theory [17] proposed by economist Schumpeter are the fundamental theories of related research. Based on Schumpeter's capital allocation efficiency theory, Wurger used an input–output efficiency method to evaluate the capital allocation efficiency and showed that optimal capital allocation efficiency can achieve the same marginal capital efficiency and Pareto optimization [18]. Solow introduced the factor of technological progress into the economic growth model and calculated the total factor productivity [19]. Lipsey decomposed efficiency from the perspectives of engineering, technology, and economy [20]. Work by Rousseau researched the efficiency of investment in financial science and technology [18]; Hartmann and Timmer used DEA models to measure input and output efficiency and evaluate enterprise science and technology input efficiency [21]. Console analysis has documented that technological change represented by information technology positively affects the efficiency of financial institutions [22].

Research on financial technology in China only began recently, and most research focuses on the impact of financial technology on macroeconomic analysis. Zhao and others

first studied and defined the relevant theories of financial technology [23]. Yu demonstrated that scientific and technological innovation had a positive effect on promoting financial investment [24]. Wang et al. tested an analytic hierarchy process (AHP) to build a capital input and technology output model to evaluate the efficiency of integrating science, technology, and finance [25]. Based on panel data of many provinces and cities, Xu and others used the Malmquist index to analyze the technical efficiency [26]. Du used a three-stage DEA model to study financial science and technology's input and output efficiency in various provinces and cities [27]. Wang applied a DEA-BCC model to build the evaluation system of enterprise's scientific and technological innovation input–output efficiency and measures the efficiency of enterprise's financial resource allocation [28].

This paper applies the DEA-Malmquist index method [29] supplemented by both a distance function and time to compare the dynamic changes in productivity of Chinese Commercial Banks. We investigate banks with the highest comprehensive technical efficiency and study their digital maturity and digital transformation experience from the perspectives of strategy and organization, products and services, talent system, infrastructure and technology layout, and risk management and control. Our approach then combines this evaluation with theoretical research and practical experience of digital transformation in China and abroad. It then presents a path forward for banks to adopt to advance their digital transformation.

The structure of this paper is as follows: the first section describes research background and literature review; the second section introduces the construction of the theoretical model; the third section discusses the data sources and indicators; the fourth section measures the financial technology efficiency of commercial banks under the benchmark parameters and makes an empirical analysis; the fifth section summarizes the full text and gives suggestions on the path of digital transformation.

## 2. Theory and Related Models

### 2.1. A Qualitative Model: Conceptual, Analytical Framework

Fintech innovations affect commercial banks and their total factor productivity through disruptive technologies. For example, consumer credit has been growing in recent years. However, due to the lack of an accurate personal information database, consumer credit is considerably constrained. The application of machine learning technology can satisfy and expand this individual credit demand [30]. Fintech can also contribute to the local economy [31]. Figure 1 shows that the development of fintech and the digitalization of commercial banks through investment in science and technology can generate a substantial impact on sustainable productivity improvement.

In Figure 1, commercial banks initiate their investments in fintech and digitalization activities for business transformation and efficiency improvement, which then triggers total factor productivity improvements through three main channels. The first is the production channel, through which more efficient production technologies generate savings and lead to improved production efficiency. Second is the transaction channel, through which commercial banks' transaction costs such as customer searching, transaction negotiation, and contract implementation are lowered through the adoption of fintech and digitalization toolkits. Third is the management channel, through which the internal management efficiency is advanced. The improvement of total factor productivity will boost the technology's efficiency and scale efficiency for commercial banks. As a result, the achievement of total factor productivity improvement provides incentives for commercial banks to invest more in science and technology, leading to a positive feedback mechanism.

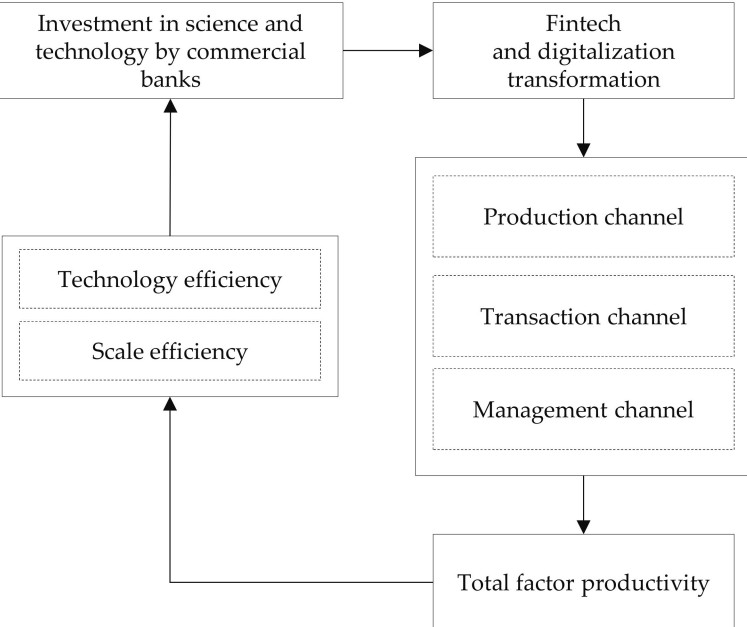

**Figure 1.** An analytical framework for commercial bank productivity improvement.

The model highlights the three channels through which bank's technology investment, particularly fintech and digitalization transformation, can improve their efficiency indexes such as TFP. We apply the DEA model to verify the theoretical model and corresponding hypothesis that commercial banks improve their productivity, efficiency, and performance through technology investment and fintech and digitalization transformation. It is expected that investment in science and technology improved the productivity of commercial banks directly; further, we hypothesize that our DEA and cluster analysis will reveal that fintech and digitalization indirectly enhance the overall technology contribution on TFP and other productivity indexes as a whole.

### 2.2. A Quantitative Model: Data Envelopment Analysis (DEA)

Data envelopment analysis (DEA) is an evaluation method based on relative efficiency. Fare and Grosskopf [32] and Zhang [33] apply the DEA to measure the estimators of the total factor productivity (TFP) index and changes in technological efficiency and scale efficiency. Using input–output indicators, the DEA method determines the effective production frontier and identifies the distance between the effective production frontier and the observed economic variable, enabling the efficiency value of the observed economic variable, the decision-making unit (DMU) to be computed. Based on the distance function concept, the Malmquist index model measures the total factor productivity (TFP), and the DEA-Malmquist index model is a nonparametric method to estimate TFP, that is, the Malmquist productivity index. When the Malmquist productivity index is greater (less) than 1, TFP has a positive (negative) growth rate from period $t$ to $t + 1$.

Against the backdrop of the digital transformation of commercial banks in China, this paper aims to evaluate the sustainable technology input efficiency of commercial banks through the adoption of DEA-Malmquist index model to estimate the changes of efficiencies of commercial banks. The model considers the changes in productivity in different periods. The index models of variable returns to scale, input and output orientation, are expressed as follows:

$$\text{M}(y_{t+1}, x_{t+1}, y_1, x_1) = \sqrt{\frac{D^t(x_{t+1}, y_{t+1}) * D^{t+1}(x_{t+1}, y_{t+1})}{D^t(x_t, y_t) * D^{t+1}(x_t, y_t)}}, \tag{1}$$

where $x$ refers to the input elements, y refers to the output elements, and D $(x, y)$ refers to the independent evaluation unit. M $(y_{t+1}, x_{t+1}, y_1, x_1)$ is the Malmquist index and represents

the change in total factor productivity (TFP) from *t* to *t + 1*. If M is greater (less) than 1, it means that the production efficiency of the unit increases (decreases) from *t* to *t + 1*. $D^t(x_{t+1},y_{t+1})$ indicates the efficiency level of phase *t + 1* under the technical factors at time *t*, and $D^t(x_t,y_t)$ represents the efficiency level of the current period under the technical factors at time t. Using the DEA model, we calculate four indexes ($D^{t+1}(x_{t+1},y_{t+1})$, $D^t(x_t,y_t)$, $D^t(x_{t+1},y_{t+1})$, and $D^{t+1}(x_t,y_t)$, respectively, in Model (1). Applying linear programming, the specific model is as follows:

$$\left[D^t(x_{t+1},y_{t+1})\right]^{-1} = max_{\phi\lambda}\phi, \; s.t. - \phi y_{i,t+1} + y_{0,t}\lambda \geq 0, \; x_{i,t+1} - x_{0,t}\lambda \geq 0 \tag{2}$$

$$\left[D^{t+1}(x_t,y_t)\right]^{-1} = max_{\phi\lambda}\phi \; s.t. - \phi y_{i,t} + y_{0,t+1}\lambda \geq 0, \; x_{i,t} - x_{0,t+1}\lambda \geq 0 \tag{3}$$

At the same time, the production efficiency can be divided into scale efficiency and technical efficiency, which are measured separately.

$$M_0 = effch * techch = pech * sec * techch \tag{4}$$

where *effch* represents the improvement of production efficiency and measures the relative efficiency of each evaluated unit improvement from time *t* to time *t + 1*. *Techch* refers to the technology upgrading, and *pech* to the efficiency of the pure technical factors, and *sech* refers to the change of scale efficiency.

## 3. Data Sources and Index Selection

### 3.1. Data Sources

By the end of 2020, China had more than 600 commercial banks (including rural commercial banks and credit unions). Due to significant differences in scale and technology investment, many Chinese banks are in different stages in their demand for digitization. This paper analyzes the scale indicators of commercial banks (including asset scale and growth rate, deposit and loan scale and growth rate, and deposit and loan proportion); performance indicators (including operating revenue and growth rate, profit and growth rate, weighted rate, and cost–income ratio); net interest margin and intermediate business income proportion; asset quality and capital indicators (non-performing rate of asset quality, loan allocation ratio, and capital adequacy ratio); and employee income ratio. Based on data availability, 50 commercial banks are selected as the research objects, including ICBC, Bank of Communications, China Construction Bank, Agricultural Bank of China, Bank of China, Bank of Beijing and Everbright Bank of China, Huaxia Bank, Minsheng Bank, Bank of Nanjing, Bank of Ningbo, Ping An Bank, Shanghai Pudong Development Bank, industrial bank, China Merchants Bank and China CITIC Bank, postal savings bank of China, Bank of Shanghai, Bank of Jiangsu, Zheshang Bank, Bohai bank, Hengfeng bank, Xiamen International Bank, Shanghai Rural Commercial Bank, Chongqing rural commercial bank, Bank of Hangzhou and Beijing Rural Commercial Bank of Guangzhou, Bank of Changsha, Bank of Chengdu, Bank of Guiyang, Bank of Jilin, Bank of Dalian, Bank of Zhengzhou, Jiangnan Rural Commercial Bank, Bank of Lanzhou, Bank of Dongguan, Bank of Hankou, Bank of Hebei, Bank of Chang'an, Bank of Hubei, Bank of Kunlun, Bank of Qingdao, Bank of Suzhou, Tianjin Rural Commercial Bank, Bank of Guilin, Qingnong commercial bank, etc. This paper uses a panel of these 50 commercial banks from 2011 to 2019, where the data are from the CSMAR database and the annual reports of commercial banks. The Appendix A lists further details of these banks including their assets, profits and non-performing loans (NPL).

### 3.2. Index Selection

Based on the availability of data, the input indexes of financial technology sustainability of commercial banks are selected: the transaction amount of Digital Banking Channel

($X_1$), the number of current digital banking channel transaction customers ($X_2$), and the number of scientific and technological staff ($X_3$) [34,35].

All three indexes represent the bank's investment in technology. For the output indicators, the revenue generated by digital banking channel ($Y_1$) and profit generated by digital banking channel ($Y_2$) are selected. These two indicators represent the output of technology factors of banks. It is noteworthy that the lack of financial science and technology talents is one of the challenges in the digital transformation faced by banks. Talent reserve construction and on-the-job staff training transformation are important for digital transformation, so we apply scientific and technological personnel as evaluation indicators. The summary of relevant inputs and outputs is shown in Table 1.

**Table 1.** Evaluation index system of financial technology allocation efficiency of banks.

| Factors | Input-Output Variables | Meaning |
|---|---|---|
| | Evaluation index | Meaning of indicators |
| Input indexes | Transaction amount of Digital Banking Channel ($X_1$) | Sustainable science and technology investment balance of invalid decision unit |
| | Number of current digital banking channel transaction customers ($X_2$) | Number of customers transacted through digital banking channels |
| | Number of scientific and technological staff ($X_3$) | Number of scientific and technological staff of commercial banks |
| Output indicators | Revenue from digital banking channels ($Y_1$) | Income generated by digital banking channels of commercial banks |
| | Profit generated by Digital Banking Channel ($Y_2$) | Profits generated by digital banking channels of commercial banks |

## 4. Empirical Analysis Results

Based on the historical bank data, the Malmquist index represents the change in total factor productivity and dynamically measures the continuous evolution of financial technology investment efficiency. In the empirical analysis, we find that the difference in science and technology investment efficiency of banks affects their comprehensive efficiency and income. The details are as follows.

### 4.1. Basic Conditions and Assumptions of the Project

Table 2 presents the total factor productivity and decomposition of financial technology of commercial banks.

Table 2 shows that from 2011 to 2019, the average TFP growth rate of financial technology of banks is 10.7%. The changes in growth rates indicate that the investment efficiency of financial technology has risen rapidly during the nine years, and Figure 2 shows the changes in TFP per year. The technological progress change index and technical efficiency change index grew the fastest, reaching 8.4% and 2.2%, respectively. The pure technical efficiency change index was 1.1%, and the scale efficiency change index was only 1%. A likely explanation is that banks in China are hindered by relatively small-scale operations and market volume that hinder the improvement of scale efficiency. Therefore, the implication for banks is that they should strengthen the management of various resources, improve risk management ability, constantly carry out financial innovation, open up new business fields, and improve the utilization efficiency of financial resources of China's banks [36]. Recently, the investment efficiency of financial technology sustainability of banks substantially increased by 25.2% from 2018 to 2019. A driving factor responsible for the progress of financial technology is the in-depth applications of cloud computing, big data, blockchain [37], and other technologies, as well as the increase in the proportion of technical staff in banks. The new round of development and application of information technology represented by big data, cloud computing, blockchain, and artificial intelligence

has created new business opportunities for banks. Driven by the transformation of financial business forms, the ecological pattern of the financial industry has also changed [38]. Fintech also reduces the credit risk of banks [39] and increases their productivity and market performance.

**Table 2.** Total factor productivity indexes and decomposition of financial technology of commercial banks.

| Time | Technical Efficiency Change | Technology Progress Change | Change Index of Pure Technical Efficiency | Scale Efficiency Change | Total Factor Productivity |
|---|---|---|---|---|---|
| | *effch* | *techch* | *pech* | *ech* | *tfpch* |
| 2011–2012 | 0.988 | 1.161 | 1.01 | 0.978 | 1.148 |
| 2012–2013 | 1.001 | 1.091 | 0.984 | 1.017 | 1.092 |
| 2013–2014 | 1.019 | 1.116 | 1.023 | 0.997 | 1.138 |
| 2014–2015 | 1.019 | 1.057 | 1.013 | 1.006 | 1.077 |
| 2015–2016 | 0.983 | 0.993 | 0.996 | 0.987 | 0.975 |
| 2016–2017 | 1.029 | 0.979 | 1.014 | 1.015 | 1.007 |
| 2017–2018 | 1.007 | 1.187 | 0.989 | 1.018 | 1.195 |
| 2018–2019 | 1.134 | 1.104 | 1.061 | 1.068 | 1.252 |
| Mean | 1.022 | 1.084 | 1.011 | 1.010 | 1.107 |

According to a report by the International Data Corporation (IDC), China's banking industry's overall IT investment scale in 2018 was 112.1 billion yuan and is projected to continue to grow over the next several years. Figure 1 plots the efficiency change of commercial banks from 2011 to 2019.

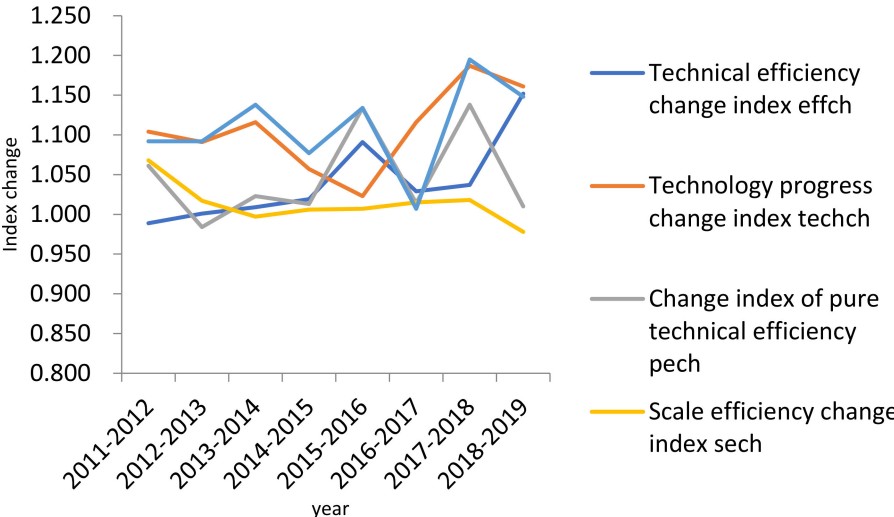

**Figure 2.** Analysis of micro–efficiency of commercial banks from 2011 to 2019.

*4.2. Analysis of Dynamic Efficiency Change of Commercial Banks*

This paper estimates a DEA-Malmquist index and each decomposition index for all 50 listed commercial banks from 2011 to 2019; using this efficiency index, we can evaluate the dynamic trend of financial technology investment of commercial banks.

Table 3 shows that the average TFP index of financial technology investment of 50 commercial banks is 1.0627. The TFP index of 40 commercial banks such as the Industrial and Commercial Bank of China is higher than the average level, which implies that the overall efficiency of financial technology investment is higher. The growth rate of these fourteen banks exceeds 10%, indicating a significant increase in financial technical efficiency. The relatively large efficiency improvement signals that banks are rapidly implementing financial technology.

From the perspective of technological progress and efficiency, China Construction Bank, Bank of Beijing, China Everbright Bank, Huaxia Bank, Minsheng Bank, Bank of Ningbo, Ping An Bank, Shanghai Pudong Development Bank, and ten other commercial banks have improved technological progress and efficiency and the overall efficiency of financial science and technology investment.

At the same time, 50 commercial banks have exhibited strong technological progress, which may be driven by the government's policy of promoting science and technology. In terms of our estimated change index of pure technical efficiency, ten banks have a Malmquist index of ten banks less than one. In contrast, the remaining forty banks' indices exceed one, implying that technical factors play an important role in improving bank efficiency [40,41]. The emergence of advanced technologies such as big cloud computing, big data, and blockchain have likely contributed to these gains by saved costs, improving efficiency and promoting the transformation from scale economy to digital economy.

We then grouped banks by TSP indices, which reflect not only the contribution of technology investment but also the overall contribution of technology through fintech and digitalization transformation as shown in Table 4 and Figure 3. We then grouped banks by TSP indices, which reflect not only the contribution of technology investment but also the overall contribution of technology through fintech and digitalization transformation as shown in Table 4 and Figure 3.

**Table 3.** Malmquist indexes of commercial banks.

| Serial Number | Bank | Technical Efficiency Change Index (effch) | Technology Progress Change Index (techch) | Change Index of Pure Technical Efficiency (pech) | Scale Efficiency Change Index (sech) | Total Factor Productivity Index (tfpch) |
|---|---|---|---|---|---|---|
| 1 | Industrial and Commercial Bank of China | 1.0000 | 1.0560 | 1.1300 | 1.1900 | 1.1510 |
| 2 | the Agricultural Bank of China | 0.9930 | 1.1320 | 0.9940 | 0.9990 | 1.1490 |
| 3 | China Construction Bank | 1.1220 | 1.0880 | 1.0000 | 1.1220 | 1.1400 |
| 4 | Bank of China | 0.9910 | 1.0950 | 1.0260 | 0.9660 | 1.1370 |
| 5 | Postal Savings Bank of China | 0.9730 | 1.1040 | 0.9890 | 0.9840 | 1.1290 |
| 6 | Bank of Communications | 1.0440 | 1.0810 | 1.0000 | 1.0440 | 1.127 |
| 7 | China Merchants Bank | 1.0470 | 1.0890 | 1.0500 | 0.9980 | 1.127 |
| 8 | Industrial Bank | 1.069 | 1.054 | 1.067 | 1.002 | 1.1270 |
| 9 | China CITIC Bank | 1.069 | 1.054 | 1.067 | 1.002 | 1.1270 |
| 10 | China Minsheng Bank | 0.991 | 1.082 | 1.00 | 0.991 | 1.1250 |
| 11 | Shanghai Pudong Development Bank | 1.0690 | 1.0540 | 1.0670 | 1.0020 | 1.1170 |
| 12 | Everbright Bank | 1.0690 | 1.0540 | 1.0670 | 1.0020 | 1.1110 |
| 13 | Ping An Bank | 0.9910 | 1.0820 | 1.1300 | 0.9910 | 1.1090 |
| 14 | Guangfa bank | 1.0820 | 1.0630 | 0.9930 | 1.0890 | 1.1090 |
| 15 | Huaxia Bank | 1.0390 | 1.0750 | 1.0280 | 1.0110 | 1.0950 |
| 16 | Bank of Beijing | 1.0290 | 1.0800 | 1.0270 | 1.0020 | 1.0890 |
| 17 | Bank of Shanghai | 0.9950 | 1.0250 | 1.1700 | 0.9950 | 1.0860 |
| 18 | Bank of Jiangsu | 1.0260 | 1.1080 | 1.0260 | 1.0450 | 1.0820 |
| 19 | Zheshang Bank | 0.9920 | 0.9870 | 0.8991 | 1.0760 | 1.0820 |
| 20 | Bank of Nanjing | 1.0230 | 0.9720 | 0.9950 | 1.2020 | 1.0820 |
| 21 | Bank of Ningbo | 1.0250 | 1.1900 | 0.9950 | 1.0200 | 1.0800 |
| 22 | Bohai Bank | 1.1080 | 1.0260 | 0.9710 | 1.1220 | 1.0800 |
| 23 | Zijin Rural Commercial Bank | 0.9870 | 0.8991 | 1.0760 | 0.9910 | 1.0800 |
| 24 | Xiamen International Bank | 0.9720 | 0.9950 | 1.2020 | 0.9730 | 1.0750 |
| 25 | Bank of Ningxia | 1.1040 | 0.9890 | 1.0880 | 1.0440 | 1.0750 |

**Table 3.** *Cont.*

| Serial Number | Bank | Technical Efficiency Change Index (effch) | Technology Progress Change Index (techch) | Change Index of Pure Technical Efficiency (pech) | Scale Efficiency Change Index (sech) | Total Factor Productivity Index (tfpch) |
|---|---|---|---|---|---|---|
| 26 | Bank of Ningxia | 1.0810 | 1.0200 | 1.0950 | 1.0260 | 1.0750 |
| 27 | Bank of Hangzhou | 0.9720 | 0.9950 | 1.1040 | 0.9890 | 1.0740 |
| 28 | Beijing Rural Commercial Bank | 1.0000 | 0.9950 | 1.0820 | 1.0800 | 1.072 |
| 29 | Bank of Guangzhou | 1.0260 | 1.0800 | 1.0630 | 1.1090 | 1.0720 |
| 30 | Bank of Changsha | 0.8991 | 1.0760 | 1.0750 | 0.9910 | 1.0630 |
| 31 | Bank of Chengdu | 1.0000 | 1.1370 | 1.0800 | 1.0820 | 1.0630 |
| 32 | Guiyang Bank | 1.0760 | 0.9450 | 1.0250 | 1.0390 | 1.0560 |
| 33 | Shenzhen Rural Commercial Bank | 1.2020 | 1.1090 | 1.1270 | 1.0290 | 1.0290 |
| 34 | Jilin Bank | 1.0880 | 1.0900 | 1.1270 | 1.1040 | 1.0280 |
| 35 | Bank of Dalian | 1.0950 | 1.0260 | 1.0720 | 1.0750 | 1.0270 |
| 36 | Bank of Zhengzhou | 1.1040 | 0.9890 | 1.0560 | 1.0800 | 1.0270 |
| 37 | Jiangnan Rural Commercial Bank | 1.0260 | 1.0090 | 1.1320 | 1.0250 | 1.0260 |
| 38 | Bank of Lanzhou | 0.8991 | 1.0760 | 1.0880 | 1.1080 | 1.0200 |
| 39 | Dongguan Bank | 1.0390 | 1.0750 | 1.0950 | 0.9870 | 1.0110 |
| 40 | HanKou Bank | 1.0290 | 1.0800 | 1.0820 | 0.9890 | 1.0020 |
| 41 | Bank of Hebei | 0.9950 | 1.0250 | 1.0630 | 1.0800 | 0.9950 |
| 42 | Changan Bank | 1.0260 | 1.1080 | 1.0750 | 1.1090 | 0.9910 |
| 43 | Bank of Hubei | 0.9920 | 0.9870 | 1.0800 | 1.0270 | 0.9910 |
| 44 | Kunlun Bank | 0.9890 | 1.0880 | 1.0250 | 1.0390 | 0.9890 |
| 45 | Bank of Qingdao | 1.0923 | 1.0950 | 1.1080 | 1.0040 | 0.9870 |
| 46 | Bank of Suzhou | 0.9950 | 1.1040 | 0.9950 | 1.0820 | 0.9840 |
| 47 | Tianjin Rural Commercial Bank | 0.9950 | 1.0820 | 1.0000 | 1.0630 | 0.9780 |
| 48 | Guilin Bank | 1.0120 | 1.0630 | 1.0760 | 1.0750 | 0.9720 |
| 49 | Qingnong Commercial Bank | 1.0880 | 1.1010 | 1.1040 | 0.9950 | 0.9660 |
| 50 | Shunde Rural Commercial Bank | 1.0950 | 1.0260 | 1.0820 | 1.0300 | 0.9450 |
| | **Mean** | 1.0325 | 1.0563 | 1.0594 | 1.0416 | 1.0627 |

Note: the data in Table 3 are the average number for the research period.

**Table 4.** Summary of basic information of clustering categories.

| Clustering Category | Frequency | Percentage (%) |
|---|---|---|
| Cluster 1: Leading Group | 17 | 33.33% |
| Cluster 2: Following Group | 2 | 3.92% |
| Cluster 3: Ordinary Group | 20 | 39.22% |
| Cluster 4: Backward Group | 12 | 23.53% |
| total | 51 | 100% |

Table 4 and Figure 3 show that there are four clusters where Cluster 1 is the leading group with the highest efficiency, followed by relatively high efficiency in Cluster 2. Cluster 3 possesses average efficiency, and Cluster 4 possesses low overall efficiency and is substantially below the production frontier. The grouping of other efficiency indexes is reported in Tables 5 and 6 and Figure 4.

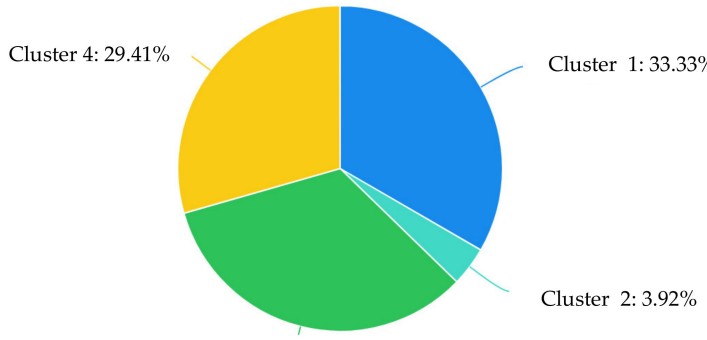

**Figure 3.** Cluster category summary.

**Table 5.** Comparison results of variance analysis of cluster categories.

| | Comparison Results of Variance Analysis of Cluster Categories (Mean ± Standard Deviation) | | | | *F* | *p* |
|---|---|---|---|---|---|---|
| | Cluster_1 (*n* = 17) | Cluster_2 (*n* = 2) | Cluster_3 (*n* = 20) | Cluster_4 (*n* = 12) | | |
| Technical efficiency change index | 1.04 ± 0.04 | 0.90 ± 0.00 | 1.05 ± 0.06 | 1.01 ± 0.03 | 7.433 | 0.000 ** |
| Technology progress change index | 1.09 ± 0.03 | 1.08 ± 0.00 | 1.03 ± 0.06 | 1.06 ± 0.05 | 5.529 | 0.002 ** |
| Change index of pure technical efficiency | 1.03 ± 0.04 | 1.08 ± 0.01 | 1.10 ± 0.04 | 1.03 ± 0.06 | 8.835 | 0.000 ** |
| Scale efficiency change index | 1.02 ± 0.04 | 1.05 ± 0.08 | 1.03 ± 0.04 | 1.10 ± 0.05 | 11.334 | 0.000 ** |
| Total factor productivity index | 1.12 ± 0.02 | 1.04 ± 0.03 | 1.03 ± 0.04 | 1.04 ± 0.06 | 14.918 | 0.000 ** |

$*\ p < 0.05\ **\ p < 0.01.$

**Table 6.** Cluster centers.

| | Initial Cluster Center | | | | Final Cluster Center | | | |
|---|---|---|---|---|---|---|---|---|
| | Cluster_1 | Cluster_2 | Cluster_3 | Cluster_4 | Cluster_1 | Cluster_2 | Cluster_3 | Cluster_4 |
| Technical efficiency change index | 1.479 | −2.327 | 0.029 | 0.400 | 0.081 | −2.415 | 0.380 | −0.346 |
| Technology progress change index | 0.071 | −0.605 | −0.591 | 1.410 | 0.586 | 0.372 | −0.585 | 0.082 |
| Change index of pure technical efficiency | 1.200 | 2.029 | 1.285 | −0.651 | −0.516 | 0.402 | 0.707 | −0.515 |
| Scale efficiency change index | 0.784 | 0.891 | 0.582 | 2.883 | −0.479 | 0.147 | −0.287 | 1.133 |
| Total factor productivity index | 1.389 | −0.979 | −0.202 | −0.993 | 0.977 | −0.385 | −0.508 | −0.474 |

Sum of squares of error SSE: 157.462.

Table 5 shows that the groups are significant for all research items ($p < 0.05$). This implies that four cluster groupings possess substantial differences in characteristics. Figure 4 and Table 6 further support the conclusion that TFP significantly differs between cluster groups. Specifically, for the highest cluster, Cluster 1, commercial banks are large and possess continuous investment in technology; They focus on fintech innovations, which can be seen through their innovative product and services. Cluster 4, the backward group, contains largely small and medium-sized banks that likely cannot afford expensive investment in fintech and digitalization transformation.

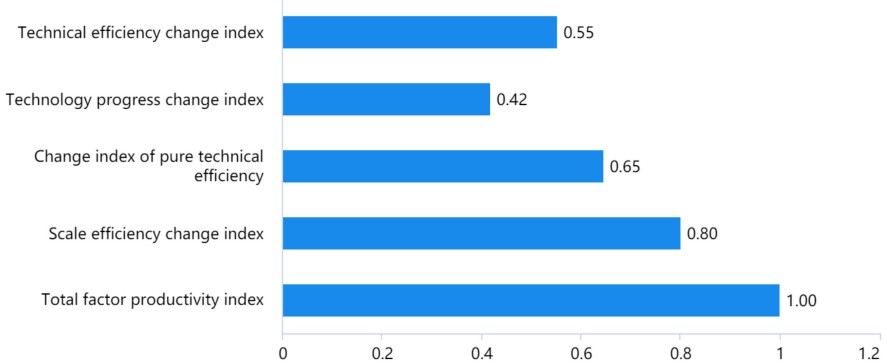

**Figure 4.** Comparison of the importance of clustering items.

Figure 5 provides visual analysis of the pure technical efficiency value and scale efficiency value of 50 commercial banks. The figure demonstrates great differences in the digital efficiency of commercial banks. ICBC has reached the forefront of production. Te Bank of Beijing, Bank of Communications, China Construction Bank, Bank of Ningbo, and Shanghai Pudong Development Bank possess high pure technical efficiency, accounting for 10% of the total number of banks. The scale efficiency of Agricultural Bank of China, Bank of China, and Huaxia Bank is also high, accounting for 10% of the total number of banks. Ping An Bank, Industrial Bank, Hankou Bank, Bank of Hebei, Chang'an bank, Bank of Hubei, Bank of Kunlun, Bank of Qingdao, Bank of Suzhou, Tianjin Rural Commercial Bank, Guilin Bank, and Qingnong have great potential to improve their pure technical efficiency and scale efficiency. Kunlun Bank's efficiency indicators are at the lowest level, and its financial technology investment is poor, indicating it should boost its business scale and operational efficiency. Bank technology investment leads to the use and development of fintech; in turn, this generates improvement of bank transaction efficiency, production efficiency, and management efficiency, thus promoting the improvement of production efficiency.

Our empirical analysis shows that the investment level of science and technology leads to the difference in comprehensive efficiency of banks and ultimately affects their economic performance and productivity [42]. Current competition among banks implies that all banks need to emphasize the importance of business innovation and productivity improvement through science and technology investment.

We chose pure technical efficiency and scale efficiency for comparative analysis. Through the comparative analysis of pure technical efficiency value and scale efficiency value, we identified the impact of technical efficiency on bank efficiency. Figure 5 highlights large differences in technical efficiency and scale efficiency between banks.

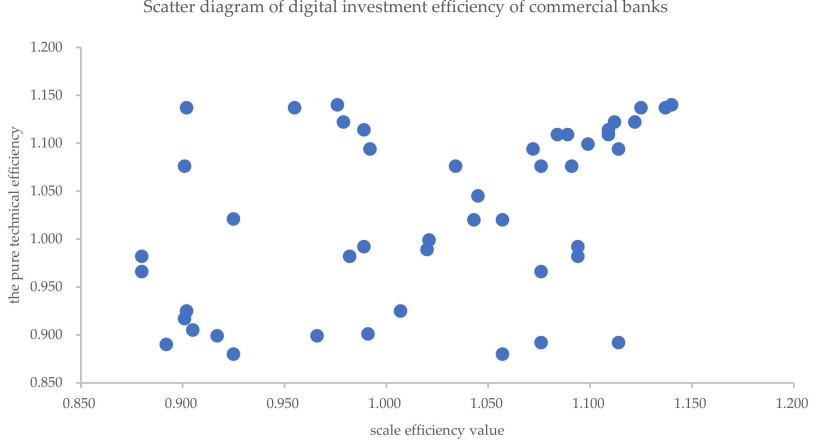

**Figure 5.** Scatter diagram of digital investment efficiency of commercial banks.

## 5. Suggestions on the Digital Transformation Path of Commercial Banks

Digital technology is gradually leading the development of commercial banks. We analyzed data collected by the China Household Finance Survey (CHFS) in 2017 and found that the mobile payment function significantly increases the possibility of family entrepreneurship and makes users take more risks, enrich social networks, and provide additional loans [43]. Thus, the digital transformation and efficiency generated through the digitalization of commercial banks have attracted widespread attention in the financial industry. In the digitalization process, it is essential to evaluate and measure the digitalization input and output performances of banks to dynamically optimize resource allocation, foster capabilities, and continuously improve commercial bank's efficiency. To test this hypothesis, we selected 50 commercial banks in China and then analyzed the impact of technology investment from 2011 to 2019 from the perspective of the technical efficiency of digital investment.

The O2O business model brought about a revolutionary impact on the marketing strategy of commercial bank retail businesses. Traditional commercial banks starting with physical business outlets are facing the saturation of the banking retail business market and the competition of the existing market by internet financial enterprises. This implies that banks need to adopt new marketing strategies to adapt to customers' new purchase behavior, enabling them to steadily develop retail business on the basis of maintaining their competitiveness and market share [44].

Furthermore, this paper offers suggestions on digital transformation and the improvement of digital investment efficiency for commercial banks. Our empirical study found that the difference in science and technology investment of commercial banks affects the comprehensive efficiency and further affects the income or revenue of banks. Through theoretical analysis, we show that bank technology investment plays an important role in the development of fintech. Fintech promotes production efficiency, transaction efficiency and management efficiency, and improves the production efficiency of banks. Figure 6 illustrates the overall path for the digital transformation for banks.

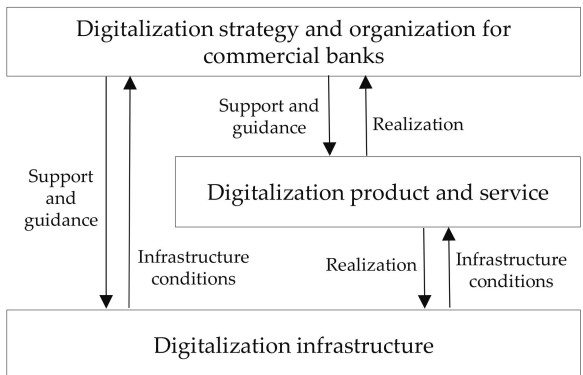

**Figure 6.** Digital transformation for commercial banks.

### 5.1. Digitalization Strategy and Organization for Commercial Banks

Digitalization strategy and organization play a decisive role for banks to dedicate resources necessary for the construction of digitalization infrastructure and provision of digital products and services. Therefore, banks need to develop medium- to long-term strategic plans for the digitalization of banks. Their digitalization strategy should not only focus on the improvement of existing products and market capabilities but also involve the construction of digitalization infrastructures. By identifying the bottlenecks of digitalization, banks can allocate financial capital, human resources, and technology capabilities to set up digital platforms, gradually upgrade products and services, effectively control financial risk, and develop innovative marketing tools.

In order to implement a digitalizaiton strategy, banks should designate these tasks to a specific organization within their organizations. For example, in the transformation process from offline business to online business, ICBC launched the strategic transformation of finance technology and smart banking ecosystem and established one digitalization department, three digital centers, one dedicated digitalization company, and one digital research institute. Hence, to ensure the smooth implementation of digitalization strategy, banks need to construct an open, collaborative, and integrated business process and supporting organizations.

### 5.2. Digitalization Infrastructure for Commercial Banks

Digitalization infrastructure is the underlying technology layout and structure for the realization of a bank's digitalization strategy and support for the digitalization transformation of products and services. Without digitalization infrastructure, it is difficult to improve efficiency and performance. Our empirical results show that technical efficiency improvement, technological progress, and pure technical factors contributed to an average total factor productivity growth of commercial banks by 10.7% over the past decade. According to the report on the construction and development process of Digital China (2019), digitalization infrastructures, including artificial intelligence, cloud computing, big data, 5G, and blockchain are at the core for the improvement of technical factors for commercial banks. Digitalization infrastructures can boost the collection of big data to facilitate customer information analysis, intelligently control potential risks, precisely implement marketing tools, and provide efficient and innovative services for the market.

### 5.3. Digitalization Product and Service for Commercial Banks

The digital transformation of commercial banks generally starts from and ends with the digitization of products and services. Our empirical analysis determines that banks with better performance always invest more resources to digitalize their products and services. Products and services such as mobile phone bank and e-loan of commercial banks are very popular (see Table 7).

**Table 7.** Representative digitalization products and services for commercial banks.

| Number | Name of Banks | Digitalization Strategy | Representative Digitalization Productcs and Services |
|:---:|:---:|:---:|:---:|
| 1 | ICBC | Digitalizaiton Bank Strategy | Mobile Phone Bank, ICBC Finance E-bank |
| 2 | CCB | Digital Bank Eco-system Strategy | Mobile Phone Bank, Dragon Pay, CCB Bank |
| 3 | BOC | Technology Leading Digitalizaiton Development Strategy | Mobile Phone Bank, BOC 5G Intellegence + |
| 4 | ABC | "iABC" Digitalization Strategy | Mobile Phone Bank, ABC e-loan, ABC Wisdom + |
| 5 | CMB | Finance Technology Bank Strategy | Mobile Phone Bank, Palm Life, U-bank X |

Commercial banks will face formidable risks in the digital transformation process to develop innovative products and services. To mitigate these uncertainties, banks can boost their product attractions and market images through innovation of digitalization products and services. Further, banks can enjoy economies of scale and scope through the substitution of traditional businesses with digitalization ones and expansion into new market niches.

In terms of policy, on 6 July 2021, the Financial Stability and Development Commission of the State Council held a meeting to strengthen the research on major topics such as digital finance and advocated the development of fintech and extensive investment in science and technology by commercial banks. Their suggestions are consistent with the

current direction of finance and its links with technology [45]. The driving forces behind it are the rapid development of digital data and digital platforms.

This paper investigates commercial banks with the highest comprehensive technical efficiency. We show that their digital maturity and digital transformation experience from the perspectives of strategy and organization, products and services, talent systems, infrastructure and technology layout, and risk management and control provides a reference for other banks to promote their digital transformation and efficiency improvement.

**Author Contributions:** All authors have contributed to the designed theoretical model, data collection, and econometric analysis and write up the paper. All authors have read and agreed to the published version of the manuscript. L.Z. (Lihua Zuo) collected the data and literature review; worked on the model, overall theme of the paper, and much of the writing; J.S. checked the model and also worked on writing the paper; L.Z. (Lijuan Zuo) helped design the theoretical model.

**Funding:** The support by the National Social Science Foundation research project "the Realization Path of Supporting the Comprehensive Construction of a Modern Country with Strong Transportation Network" (21AZD019) is gratefully acknowledged.

**Institutional Review Board Statement:** Not applicable.

**Informed Consent Statement:** Not applicable.

**Data Availability Statement:** Financial data for commercial banks can be downloaded at https://www.gtarsc.com/ (accessed on 15 September 2021), and the annual reports for commercial banks can be found at https://money.163.com/ (accessed on 15 September 2021).

**Conflicts of Interest:** The authors declare that they have no conflict of interest with any other parties.

**Appendix A**

Table A1. Data summary of 50 commercial banks.

| Serial Number | Organization Name | Net Core Tier 1 Capital (RMB 100 Million) (2019) | Asset Scale (RMB 100 Million) (2019) | Net Profit (RMB 100 Million) (2019) | Cost Income Ratio (%) (2019) | NPL Ratio (%) (2019) |
|---|---|---|---|---|---|---|
| 1 | Industrial and Commercial Bank of China | 22,320.33 | 276,995.4 | 2987.23 | 23.91 | 1.52 |
| 2 | the Agricultural Bank of China | 158,3927 | 226,094.71 | 2026.31 | 31.27 | 1.59 |
| 3 | China Construction Bank | 1,889,390 | 232,226.93 | 2556.26 | 26.42 | 1.59 |
| 4 | Bank of China | 14,657.69 | 212,672.75 | 1924.35 | 28.09 | 1.42 |
| 5 | Postal Savings Bank of China | 4216.78 | 95,162.11 | 523.84 | 57.6 | 0.86 |
| 6 | Bank of Communications | 6348.07 | 95,311.71 | 736.3 | 31.5 | 1.49 |
| 7 | China Merchants Bank | 4823.4 | 67,457.29 | 808.19 | 31.02 | 1.36 |
| 8 | Industrial Bank | 4403.65 | 67,116.57 | 612.45 | 26.89 | 1.57 |
| 9 | China CITIC Bank | 4033.54 | 60,667.14 | 453.76 | 30.57 | 1.77 |
| 10 | China Minsheng Bank | 4157.26 | 59,948.22 | 503.3 | 30.07 | 1.76 |
| 11 | Shanghai Pudong Development Bank | 4351.2 | 62,896.06 | 565.15 | 25.22 | 1.92 |
| 12 | Everbright Bank | 2896.38 | 43,573.32 | 337.21 | 28.79 | 1.59 |
| 13 | Ping An Bank | 1997.82 | 34,185.92 | 248.18 | 30.32 | 1.75 |
| 14 | Guangfa Bank | 1568.48 | 23,608.5 | 106.99 | 36.18 | 1.45 |
| 15 | Huaxia Bank | 1981.97 | 26,805.8 | 209.86 | 32.58 | 1.85 |
| 16 | Bank of Beijing | 1757.14 | 25,728.65 | 201.37 | 25.19 | 1.46 |

**Table A1.** *Cont.*

| Serial Number | Organization Name | Net Core Tier 1 Capital (RMB 100 Million) (2019) | Asset Scale (RMB 100 Million) (2019) | Net Profit (RMB 100 Million) (2019) | Cost Income Ratio (%) (2019) | NPL Ratio (%) (2019) |
|---|---|---|---|---|---|---|
| 17 | Bank of Shanghai | 1568.48 | 23,608.5 | 106.99 | 36.18 | 1.45 |
| 18 | Bank of Jiangsu | 1038.87 | 19,258.23 | 132.63 | 26.68 | 1.39 |
| 19 | Zheshang Bank | 870.44 | 16,466.95 | 115.6 | 29.99 | 1.2 |
| 20 | Bank of Nanjing | 679.16 | 12,432.69 | 111.88 | 28.61 | 0.89 |
| 21 | Bank of Ningbo | 658.04 | 11,164.23 | 112.21 | 34.44 | 0.78 |
| 22 | Bohai Bank | 557.36 | 10,344.51 | 70.8 | 35.46 | 1.84 |
| 23 | Zijin Rural Commercial Bank | 122.45 | 1931.65 | 12.54 | 33.42 | 1.69 |
| 24 | Xiamen International Bank | 462.36 | 8061.05 | 58.24 | 25.55 | 0.73 |
| 25 | Bank of Ningxia | 116.61 | 1447.62 | 5.75 | 35.49 | 3.79 |
| 26 | Bank of Ningxia | 708.86 | 9506.18 | 91.64 | 30.35 | 1.29 |
| 27 | Bank of Hangzhou | 470.6 | 9210.56 | 54.12 | 29.91 | 1.45 |
| 28 | Beijing Rural Commercial Bank | 514.22 | 8811.28 | 72.52 | 33.53 | 0.36 |
| 29 | Bank of Guangzhou | 377.55 | 5136.2 | 37.69 | 29.76 | 0.86 |
| 30 | Bank of Changsha | 311.18 | 5266.3 | 45.78 | 34.12 | 1.29 |
| 31 | Bank of Chengdu | 313.16 | 4922.85 | 46.54 | 25.77 | 1.54 |
| 32 | Guiyang Bank | 304.72 | 5033.26 | 52.29 | 26.73 | 1.35 |
| 33 | Shenzhen Rural Commercial Bank | 269.86 | 3168.97 | 43.23 | 29.93 | 1.14 |
| 34 | Jilin Bank | 247.33 | 3618.52 | 11.57 | 41.72 | 2.82 |
| 35 | Bank of Dalian | 263.8 | 4185.73 | 16.31 | 40.03 | 2.29 |
| 36 | Bank of Zhengzhou | 287.12 | 4661.42 | 31.01 | 27.96 | 2.47 |
| 37 | Jiangnan Rural Commercial Bank | 224.41 | 3797.96 | 24.2 | 28.78 | 1.61 |
| 38 | Bank of Lanzhou | 202.84 | 3039.02 | 22.66 | 33.29 | 2.25 |
| 39 | Dongguan Bank | 205.67 | 3144.99 | 24.61 | 32.41 | 1.39 |
| 40 | HanKou Bank | 200.12 | 3192.96 | 18.82 | 34.04 | 2.11 |
| 41 | Bank of Hebei | 259.29 | 3422.43 | 20.22 | 36.15 | 2.53 |
| 42 | Changan Bank | 147.49 | 2412.57 | 15.48 | 34.83 | 1.78 |
| 43 | Bank of Hubei | 206.63 | 2424.79 | 17.52 | 25.53 | 2.21 |
| 44 | Kunlun Bank | 303.34 | 3511.38 | 32.75 | 28.21 | 1.36 |
| 45 | Bank of Qingdao | 192.69 | 3176.59 | 20.43 | 32.97 | 1.68 |
| 46 | Bank of Suzhou | 240.31 | 3110.86 | 23.14 | 37.73 | 1.68 |
| 47 | Tianjin Rural Commercial Bank | 248.07 | 3172.56 | 24.44 | 31.19 | 2.47 |
| 48 | Guilin Bank | 178.59 | 2672.88 | 16.26 | 31.61 | 1.74 |
| 49 | Qingnong commercial Bank | 206.67 | 2941.41 | 24.44 | 32.23 | 1.57 |
| 50 | Shunde Rural Commercial Bank | 271.27 | 3032.08 | 31.98 | 31.26 | 1.27 |

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
