# Peer review of "The Digitalization Transformation of Commercial Banks and Its Impact on Sustainable Efficiency Improvements through Investment in Science and Technology"

_sustainability, doi:10.3390/su131911028_

Round 1

Reviewer 1 Report

Dear authors,

Firs of all, I congratulate you for having chosen a very interesting (and hot topic) about the digitalization of the banking system.

I consider that the proposed paper is very well written, and there is in it very good material for an excellent paper.

However, from my point of view, some things need to be more developed, and the structure of the paper (basically its research logic and theoretical support) needs to be improved.

First, it is difficult for me to consider digital transformation of banks without speaking about fintech business. And, to speak about fintech is to speak about entirely new business models for banking. These new business models are at the origin of trully digital transformation processes in banking. Surprisingly, authors never refer to these topics and, despite this is briefly considered in figure 3 (the strategic conditions for digital transformation). Also, variables used do not consider these ideas.

At the same time, I consider that this paper is presented as a (very interesting) report about the explanation for the improvements of productivity in the Chinese banking system. Authors present this idea in a very clear way.

However, I cannot ascertain a real research model underlying this paper and so, from my point of view the resulting analysis and discussion becomes poor, not profiting from the richness of the analysis that authors can perform if oriented by a true conceptual research model.  

To develop this  model, authors need to go deep in the literature review (even if the model is not developed, authors need to consider revising some of the sentences presented that are not supported in any bibliographic references which is not advisable to consider - see Section 5 of the paper, starting by figure 3). 

Research needs to consider: what do we want to explain? What have we found? What are the reasons for our findings? How can we explain the observed differences? What can we learn from our research? What are our proposals?

In summary, I consider that this can become a very interesting paper, but authors need to leave the reporting style and further develop the research model and respective theoretical support.

In the end I would like to see some theoretical-supported explanations for the observed differences, the proposal of some clustering in terms of the digital transformation process for the banks in China, and some possible considerations for the future developments for this sector, departing from the state-of-the-art analyzed by authors.

Once again, I consider that all these questions do not take any merit to your research but are important to devise its real importance and possibly to reinforce its practical and theoretical support.

Reviewer 2 Report

Content & methodology

  • Reconsider using reference [3] (and potentially [4]) in line 35; Wilhem surely had / must have had a different concept of digitization compared to how this word is understood nowadays – so while the historical context is well appreciated, the relevance might be a little weak.
  • In line 86 you make the point that your research results of the leading banks may serve as a template of digitization for other banks – this is a nice conceptual idea, and you should pick it up in your summary again (lines 293 onwards).
  • In line 162 you argue that “scientific and technological staff” is a good indicator for digitization efforts – could you perhaps briefly motivate this? In my experience, at least scientific stuff of a bank may be occupied with topics totally different to digitization, e.g. economic research, quant development, …, which would be investment in technology but necessarily in digitization.
  • In line 163+164, you’ll need to differentiate Y1 and Y2 – it’s both “Digital Banking Channel”. I think it becomes clearer in Table 1, but right here it is confusing.
  • Line 182: I find it a strange result that scale efficiency is so low since it’s often quoted as one of the main arguments in favour of digitization – a discussion/explanation would be very beneficial here.
  • In line 184 and onwards (and 221 onwards), you quote several “driving factors”: a motivation, explanation, or reference would be needed to support your statement.
  • For Figure 2 (line 224), you chose to display two dimensions of the 4 (or 5) you have in your data – a motivation for the choice would be needed.

Format & wording

  • Please check/revise wording – only minor revisions needed, e.g.
    • line 242: “concern” – perhaps “attention” or similar would be a better choice
  • Please revise spelling/formatting – only minor revisions needed, e.g.
    • line 130: “Techch” – “T” not in italics
    • Table 4: ICBC – “digitalizaiton”
  • Please check revise text coherence and meaning – only minor revisions needed, e.g.
    • throughout the manuscript: “sustainable” – please be aware the term/usage “sustainable investment” in the financial community currently has a meaning that is probably not intended by the authors, i.e. “green” or ESG-conforming investment; if this is indeed the case, I advocate for omitting the word or replace it by a less ambiguous one
    • line 137: “… demand for digital” – digital what?

Presentation

  • lines 143-157: consider a table and/or pushing the list/table to the appendix
  • Figure 1: consider omitting this figure since
    • information content is the same as in Table 2
    • graph is rather jumpy
  • If you want to retain Figure 1,
    • axis need labeling
    • “Mean” should be omitted or included as horizontal lines in graph
  • Table 3: consider
    • adding to the header the information that numbers are averages over time
    • using ordering/ranking the banks so that analysis is easier and more consistent with text
    • including a full list/table in the appendix
  • Figure 2: axis need labeling to make a better link with text (lines 224-236)

Reviewer 3 Report

The article is very interesting and the theme of digital transformation was very well combined with banking.

From the point of view of article organization/structure, I suggest that authors follow the MDPI authors' guidelines – e.g., see the reference list, as the references do not strictly follow the MDPI guidelines.

Since 2018, there has been a growing number of empirical and theoretical articles published on digital transformation, so I suggest including more references from top-tier journals on the subject in order to reinforce the theoretical body of knowledge.

Finally, I also suggest that the authors to present the contradictory, i.e., the discussion about the transition from O2O (online-offline) in banking services.

Overall, my opinion is that the authors are to be congratulated on their work.

Round 2

Reviewer 1 Report

Thank you, once again, for your paper.

I see that you took good note of previous comments, and that you have tried to comply with our proposals.

I notice both, a very interesting improvement in terms of the theoretical concept, namely including the fintech's topic, and the associated digital transformation model, having reinforced the theoretical soundness of the proposed paper.

I must say that I would expect a stronger theoretical support regarding the model's ideas and the proposed hypothesis that should have allowed a stronger support also for the paper's findings and related conclusions.

I insist on the importance of grouping the results in terms of clusters regarding the level and possible stages of Digital Transformation of the Chines banks, and I encourage you to try this step if possible.

However, I consider that the current version is already a good paper, but please revise both formats and numbering of figures.
